# Transcriptome and Metabolome Analyses Reveal Sugar and Acid Accumulation during Apricot Fruit Development

**DOI:** 10.3390/ijms242316992

**Published:** 2023-11-30

**Authors:** Ningning Gou, Chen Chen, Mengzhen Huang, Yujing Zhang, Haikun Bai, Hui Li, Lin Wang, Tana Wuyun

**Affiliations:** 1State Key Laboratory of Tree Genetics and Breeding, Research Institute of Non-Timber Forestry, Chinese Academy of Forestry, Zhengzhou 450003, China; lemonn@caf.ac.cn (N.G.); chenchenbo@caf.ac.cn (C.C.); mengzhen4524@163.com (M.H.); zhangyujing@caf.ac.cn (Y.Z.); bhk1994@163.com (H.B.); lihui19971204@163.com (H.L.); wanglin1815@163.com (L.W.); 2College of Forestry, Nanjing Forestry University, Nanjing 210037, China; 3Kernel-Apricot Engineering and Technology Research Center of State Forestry and Grassland Administration, Zhengzhou 450003, China; 4Key Laboratory of Non-Timber Forest Germplasm Enhancement and Utilization of National Forestry and Grassland Administration, Zhengzhou 450003, China

**Keywords:** apricot, fruits, sugar, acid, WGCNA, transcriptome, metabolome

## Abstract

The apricot (*Prunus armeniaca* L.) is a fruit that belongs to the Rosaceae family; it has a unique flavor and is of important economic and nutritional value. The composition and content of soluble sugars and organic acids in fruit are key factors in determining the flavor quality. However, the molecular mechanism of sugar and acid accumulation in apricots remains unclear. We measured sucrose, fructose, glucose, sorbitol, starch, malate, citric acid, titratable acid, and pH, and investigated the transcriptome profiles of three apricots (the high-sugar cultivar ‘Shushanggan’, common-sugar cultivar ‘Sungold’, and low-sugar cultivar ‘F43’) at three distinct developmental phases. The findings indicated that ‘Shushanggan’ accumulates a greater amount of sucrose, glucose, fructose, and sorbitol, and less citric acid and titratable acid, resulting in a better flavor; ‘Sungold’ mainly accumulates more sucrose and less citric acid and starch for the second flavor; and ‘F43’ mainly accumulates more titratable acid, citric acid, and starch for a lesser degree of sweetness. We investigated the DEGs associated with the starch and sucrose metabolism pathways, citrate cycle pathway, glycolysis pathway, and a handful of sugar transporter proteins, which were considered to be important regulators of sugar and acid accumulation. Additionally, an analysis of the co-expression network of weighted genes unveiled a robust correlation between the brown module and sucrose, glucose, and fructose, with *VIP* being identified as a hub gene that interacted with four sugar transporter proteins (*SLC35B3*, *SLC32A*, *SLC2A8*, and *SLC2A13*), as well as three structural genes for sugar and acid metabolism (*MUR3*, *E3.2.1.67*, and *CSLD*). Furthermore, we found some lncRNAs and miRNAs that regulate these genes. Our findings provide clues to the functional genes related to sugar metabolism, and lay the foundation for the selection and cultivation of high-sugar apricots in the future.

## 1. Introduction

Apricot (*Prunus armeniaca* L.) is an important fruit tree of the Rosaceae family, and it is loved by consumers because of its unique flavor and high nutritional value [1,2,3,4,5]. Fruit flavor-related traits are usually quantitative traits that have a complex genetic regulation mechanism, and it is difficult to grasp their genetic rules. Consequently, fruit quality cannot be improved quickly and efficiently in a short period of time, thus hindering the breeding process [6]. With the improvement of people’s standard of living, the change in consumption concepts, and the internationalization of the product market, people pay more and more attention to fruit flavor, and the sugar content in fruit has a great influence on fruit flavor [7].

The studies on the sugar metabolism of Rosaceae fruit trees mainly focus on the peach [8,9,10], apricot [11,12,13], plum [14,15,16], cherry [17,18], apple [19,20,21], pear [22,23], and loquat [24]. Sucrose, fructose, glucose, and sorbitol were found to be the main sugars accumulated in the fruits of the Roseaceae family [25]. The soluble sugars in mature apples were mainly fructose, followed by sucrose and glucose, with the least amount of sorbitol [19]. However, mature peaches contain mainly sucrose, which accounts for 40–85% of the total sugar content [9,26,27]. Mature loquat fruits were mainly composed of fructose and glucose, while sucrose was relatively stable in the growth stage [28]. Glucose and fructose were mainly accumulate during the development of sweet cherry fruit, sucrose content was low, and starch and sorbitol were not accumulate [17]. Malate and citric acids were found to be the most prevalent organic acids in Rosaceae fruits [29,30]. Therefore, studies on the metabolism of the sugars and organic acids in fruits help to elucidate the complex metabolic processes in fruits.

Sugar and acid metabolism are regulated by related enzymes and transport proteins. Veronica et al. discovered that the transcription levels of sucrose synthetase and sucrose phosphate synthetase were significantly higher after fruit color change, while the transcription levels of sorbitol dehydrogenase were not significantly different in different periods [31]. Wang et al. found a hexasaccharide transporter, *MdHT2.2*, which was highly expressed in mature apple fruits, and associated with sugar content. It was a hexasaccharide/H+ homologous transporter with glucose and fructose transport activity, whose primary function was to transport glucose and fructose from the plastidic ectodomain space to pulp cells in order to maintain a high hexose unloading capacity [32]. Wu et al. identified 45 genes related to sugar metabolism in a selected region of Asian pears, such as *glgA*, *scrK*, *INV*, sorbitan transporter, hexose transporter, etc. [33]. The content of organic acids in fruit ultimately depends on the equilibrium of acid production, decomposition, utilization, and distribution [34]. Previous studies have shown that *CS*, *ACO*, *PCK*, and *MDH1* play important roles in the metabolism of fruit acid [35,36].

Here, high-performance liquid chromatography (HPLC) was performed to examine the major sugars and organic acids in the fruits of the high-sugar cultivar ‘Shushanggan’, common-sugar cultivar ‘Sungold’, and low-sugar cultivar ‘F43’ during three distinct developmental stages and unveil disparities in the sugar and organic acid levels among the fruits of the three apricot cultivars. Additionally, we employed whole-transcriptome sequencing and real-time fluorescence assays in combination with bioinformatics to uncover key genes that accumulate at the molecular level in apricot sugar and apricot acid. This study provides a scientific basis and valuable genetic resources for improving the essential characteristics of fruit, accelerating the utilization of high-sugar varieties, breeding parent species, and breeding new varieties. It provides theoretical support for improving the high-quality development and competitiveness of the apricot industry.

## 2. Results

### 2.1. Soluble Sugar and Organic Acid Content during Three Apricot Cultivars’ Fruit Development

We determined the content of total soluble sugar (TSS) in different apricot fruits, defining TSS > 11% as a high-sugar cultivar, 8%< TSS < 10% as a common-sugar cultivar, and TSS < 3% as a low-sugar cultivar (Appendix A). In this experiment, to investigate the differences in the soluble sugar and organic acid contents among different apricot cultivars, the three representative apricot cultivars (the high-sugar cultivar ‘Shushanggan’, common-sugar cultivar ‘Sungold’, and low-sugar cultivar ‘F43’) were selected for the study at three growth and development stages (green-fruit stage, color-turning stage, and commercial-maturity stage) (Figure 1A).

A significant difference was observed in soluble sugar (glucose, sucrose, fructose, and sorbitol), organic acid (citric acid and malate), and starch in the development of the three types of apricot fruits (Appendix A). In S1, sucrose, sorbitol, and citric acid contents did not change much in the three kinds of apricot (Figure 1B,E,F). The glucose contents were 17.8 and 18.9 mg/g in ‘Shushanggan’ and ‘Sungold’, respectively, while it was 1.3 mg/g in ‘F43’ (Figure 1C). The fructose contents were 4.4, 8.2, and 0.7 mg/g in ‘Shushanggan’, ‘Sungold’, and ‘F43’, respectively (Figure 1D). The level of malate and starch contents in ‘Shushanggan’ and ‘F43’ remained relatively constant, and were higher than in ‘Sungold’ (Figure 1G,H). Sucrose, glucose, fructose, sorbitol, and citric acid in ‘Shushanggan’ showed an increasing pattern from S2 to S3, with increments of 7.1–84.4 mg/g, 20.8–36.5 mg/g, 13–19.4 mg/g, 25.8–26.5 mg/g, and 6.2–9.1 mg/g, respectively (Figure 1B–F). Nevertheless, the levels of malate and starch declined from S2 to S3 (Figure 1G,H). Sucrose in ‘Sungold’ increased from 14.8 to 51.8 mg/g, while the levels of glucose and fructose were almost unchanged in S2 and S3, and both decreased compared to S1. Sorbitol, malate, and starch contents all decreased from S1 (Figure 1B–H). The contents of sucrose, glucose, and fructose in ‘F43’ increased from S1 to S3 but were lower than those in ‘Shushanggan’ and ‘Sungold’, while the contents of citric acid, and starch in ‘F43’ were higher than those in ‘Shushanggan’, and ‘Sungold’ from S2 (Figure 1B–H). In addition, we found that titratable acid was lower in ‘Shushanggan’ than ‘Sungold’ lower than ‘F43’, and had a pH greater than ‘F43’, greater than ‘Sungold’ (Appendix A). Therefore, from S2 to S3, ‘Shushanggan’ accumulates more sucrose, glucose, fructose, and sorbitol and less titratable acid and citric acid, resulting in a better flavor, and ‘Sungold’ mainly accumulates more sucrose, and less citric acid and starch, for the second flavor, whereas ‘F43’ mainly accumulates more titratable acid, citric acid, and starch for less sweetness.

### 2.2. Transcriptome Sequencing and Gene Expression in Fruit Development of Three Apricot Cultivars

To investigate the transcriptional regulation of the accumulation of the soluble sugar, and organic acid in apricot fruit, a total of 25 samples from the three apricot cultivars ‘Shushanggan’, ‘Sungold’, and ‘F43’ were sequenced using RNA−seq during three critical developmental periods. A total of 3170040224 original reads were obtained after removing the reads compared with the ribosomes. Approximately 98.86% of clean reads containing adapter, excessive N, or many low-quality bases were retained for subsequent analysis. The percentages of Q20 and Q30 bases were 97.72% and 92.55% and above, respectively, and GC content ranged from 47.33% to 49.14%. Clean reads were compared to the reference genome GDR_Prunus_sibirica_F106 using HISAT (http://www.ccb.jhu.edu/software/hisat (accessed on 11 March 2021)). Clean reads sequenced from 25 libraries had alignment rates of 67.94%–81.62% and a chain specific ratio of 91.62%–97.28% (Appendix A).

The expressed genes in individual samples of ‘Shushanggan’, ‘Sungold’, and ‘F43’ accounted for roughly 96.7%–98.2% of the total expressed genes, and SF1 accounted for the highest proportion and SJ3 the lowest (Figure 2A). The proportions of genes distributed at the four expression levels were relatively similar in all stages between the three cultivars, with 6.7%–13.7%, 39.1%–47.0%, and 31.2%–38.6% of the genes expressed at the levels of 0 < FPKM ≤ 1, 1 < FPKM ≤ 10, and 10 < FPKM ≤ 50, respectively, and approximately 12.0%–14.8% of genes showed very high expression levels (FPKM > 50) in different samples (Figure 2B). To investigate the overall variation in differences in transcriptional dynamics in the fruit growth and development of the three apricot cultivars, 29,056 expressed genes were analyzed via hierarchical clustering and PCA (principal component analysis) based on Pearson’s correlation coefficient (Figure 2C,D). The results showed that the transcriptome expression data of the three apricot cultivars at the three developmental stages were roughly divided into three groups by the two analyses, that is, the three stages of ‘F43’ (SF1, SF2, and SF3) were one group, and the correlation was higher at different stages of the same cultivars; the green-fruits stage of ‘Shushanggan’ and ‘Sungold’ (SS1 and SJ1) was one group; and the color-turning and commercial-maturity stages of ‘Shushanggan’ and ‘Sungold’ (SS2, SS3, SJ2, and SJ3) were one group, and the correlation was higher at different stages of different cultivars (Figure 2C,D). Taken together, the above results show that the differences in transcription levels between ‘Shushanggan’, and ‘Sungold’ were mainly concentrated in S2 and S3, while ‘F43’ was different from ‘Shushanggan’ and ‘Sungold’. Furthermore, the particularity of transcription levels in the S2 and S3 stages may be the reason for the difference in sugar content between high-sugar and common−sugar apricots.

### 2.3. Identification of Sugar and Acid Metabolism Genes in Three Apricot Cultivars

A total of 19,230 genes with FPKM > 5 were examined in order to determine the genes and transcription factors associated with sugar and acid metabolism. In ‘Shushanggan’, 7270 and 5008 genes were differentially expressed in SS3 compared to SS1 and SS2, 414 genes were upregulated, and 946 genes were downregulated in SS3/SS1, SS3/SS2, and SS2/SS1. In ‘Sungold’, 7108 and 6197 genes were differentially expressed in SJ3 compared to SJ1 and SJ2, 248 genes were upregulated, and 1024 genes were downregulated in SJ3/SJ1, SJ3/SJ2, and SJ2/SJ1. In ‘F43’, 5968 and 5142 genes were differentially expressed in SF3 compared to SF1 and SF2, and 354 genes were upregulated and 596 genes were downregulated in SF3/SF1, SF3/SF2, and SF2/SF1 (Figure 3A,B). To explore the reasons for the differences in sugar content among the three cultivars, the common upregulated and downregulated genes were analyzed by comparing the periods among the cultivars, and it was found that 23 upregulated genes and 115 downregulated genes were common genes. Functional analysis of the genes revealed 62 genes involved in sugar and acid metabolism (Appendix A).

KEGG analyses were performed to understand the metabolic pathways where DEGs were involved. The top 20 pathways were selected for mapping based on *p*-value, and among these pathways, the metabolic pathway was significantly enriched, except for SF3 vs. SF1 and SF2 vs. SF1, indicating that this process has a significant impact on sugar and acid buildup in apricot fruit (Appendix A).

### 2.4. Identification of DEGs Associated with Sugar and Acid Metabolism Pathways

Sugar and acid metabolism are regulated by related enzyme genes, and differences in the expression of these genes result in disparities in the accumulation of sugar during different phases of fruit growth. In the starch and sucrose metabolism pathways, the expression levels of 8 genes were highest in ‘Shushanggan’ compared to ‘F43’ and ‘Sungold’, followed by ‘Sungold’, and the lowest expression occurred in ‘F43’, including one *E2.4.1.14* (sucrose-phosphate synthase, PaF106G0100001858.01), one *SPP* (sucrose-6-phosphatase, MTCONS_00064889), one *ENPP1_3* (ectonucleotide pyrophosphatase family member 1/3, MTCONS_00060314), one *glgA* (starch synthase, MTCONS_00008488), one *WAXY* (granule-bound starch synthase, MTCONS_00047878), one *GBE1* (1,4-alpha-glucan branching enzyme, PaF106G0700027757.01), and two *E3.2.1.2* (beta-amylase, PaF106G0200009233.01 and MTCONS_00023366). In the citrate cycle pathway, the high expression of the four *MDH1s* (malate dehydrogenase, PaF106G0800031692.01, PaF106G0100006207.01, MTCONS_00044498, and MTCONS_00044495) at the ‘Shushanggan’ color change and ripening stages compared to ‘Sungold’ and ‘F43’ at each growth and developmental period suggests that the sugar−acid ratio had a considerable effect on fruit quality. Furthermore, two sugar transporter proteins, *SLC2A8* (facilitated glucose transporter, PaF106G0300013847.01 and MTCONS_00030674), were highly expressed in ‘Shushanggan’ (Figure 4). These findings suggest that these DEGs linked to sugar and acid metabolism play a crucial role in regulating the sugar buildup of apricots.

### 2.5. Identification of Co-Expression Network and Hub Genes Related to Sugar and Acid Metabolism

We used WGCNA to investigate the relationship between the expression patterns of sucrose, glucose, fructose, sorbitol, malate, citric acid, and starch, and the gene regulatory networks linked to their accumulation. After screening for DEGs, 7993 genes obtained from transcriptome sequencing were analyzed, and these gene sets were divided into thirteen co-expression modules (Figure 5A). The results showed that the correlation coefficient of the MEbrown module with the sugar trait was the highest, including sucrose (*r* = 0.89, *p* = 3.7 × 10^−9^), glucose (*r* = 0.77, *p* = 6.2 × 10^−6^), and fructose (*r* = 0.9, *p* = 1 × 10^−9^) (Figure 5B). Interestingly, the MEbrown module was found to be highly expressed in SS3, followed by SJ3, SJ2, and SS2, and low in SS1, SJ1, SF1, SF2, and SF3, which follows the same trend of high sugar as in ‘Shushanggan’ (Figure 5C). We picked out the potential candidate genes in the MEbrown module based on the high module membership (MM) and high gene significance (GS), including 19 sugar transporter proteins, 18 transcription factors, and 85 structural genes for sugar and acid metabolism. Nevertheless, only 52 genes were screened to construct gene co-expression networks based on the weight values (top 100) and the degree of connectivity between the candidate genes. *VIP1* (MTCONS_00024580) was identified as a key hub gene with 21 connected edges, Significantly, *VIP1* mainly interacted with four sugar transporter proteins (*SLC35B3*, PaF106G0500019562.01; *SLC32A*, PaF106G0300011434.01; *SLC2A8*, MTCONS_00030674; and *SLC2A13*, MTCONS_00074442), and three structural genes for sugar and acid metabolism (*MUR3*, PaF106G0100001918.01; *E3.2.1.67*, PaF106G0600022809.01; and *CSLD*, MTCONS_00043102) (Figure 5D). The expression levels of the eight hub genes were highest in ‘Shushanggan’ compared to ‘F43’ and ‘Sungold’ (Appendix A), and it is suggested that these hub genes were positive regulators of sugar accumulation in apricot.

### 2.6. Identification of DEmRNAs and Their Corresponding lncRNAs and miRNAs Involved in Sugar and Acid Metabolism Pathways

To understand the relationship between lncRNAs and miRNAs, and sugar and acid accumulation, we further selected 42 DEmRNAs for an analysis targeting lncRNAs and miRNAs, genes involved in the starch and sucrose metabolic pathways and citrate cycle pathway, sugar transporters, and genes related to sugar and acid metabolism obtained via WGCNA analysis. We analyzed 42 kinds of DEmRNAs and their corresponding lncRNAs and miRNAs, there were a total of 80 lncRNA−DEmRNA relationship pairs, including 80 lncRNAs and 36 DEmRNAs. In addition, we identified 34 miRNA−mRNA interactions of 42 DEmRNAs, including 28 miRNAs and 34 DEmRNAs (Figure 6).

### 2.7. qRT−PCR Analysis

For the validation of the gene expression pattern of DEGs in three apricot cultivars, nine genes were selected for qRT−PCR analysis (Figure 7). UBQ was used as an inner control, and the 2^−ΔΔCT^ values were calculated, which illustrated that the expression consequences conformed to the transcriptome profiling. The correlation coefficient between the RNA−seq and qRT−PCR was >0.80 (*p* < 0.05) for most of the tested genes (8/9), indicating the reliability of RNA−seq data to reflect the abundance of transcript levels.

## 3. Discussion

Fruit quality improvement in apricot is mainly due to the improvement in sugar and acid content and their proportions [37,38]. A number of potential pathways, and genes associated with sugar and acid metabolism, have been identified in Rosaceae plants. To comprehend the molecular processes by which sugar and acid accumulate in apricot, we performed phenotypic and transcriptomic analyses to pinpoint potential genes involved in sugar and acid metabolism.

Sucrose, glucose, fructose, sorbitol, citric acid, malate, and starch are main ingredients in Rosaceae fruiter [25,29,30], and this also prompted us to study the composition of three apricot cultivars at three distinct developmental stages. In this study, from color transition to ripeness, the high-sugar cultivar ‘Shushanggan’ accumulates more sucrose, glucose, fructose, and sorbitol and less titratable acid, citric acid, and starch, resulting in a better flavor, and the common-sugar cultivar ‘Sungold’ mainly accumulates more sucrose, and less citric acid and starch for the second flavor, whereas the low-sugar cultivar ‘F43’ mainly accumulates more titratable acid, citric acid, and starch for less sweetness. This finding is inconsistent with previous studies, in which it was shown that sucrose and malate were the main components of soluble sugar and organic acid in apricot [12]. In addition, the three apricot cultivars have low sucrose content and slow growth in the early stages of fruit development (Figure 1B), which may be due to the fact that sucrose, produced via photosynthesis in the leaves, is transported to the inside of the fruit, where it breaks down into glucose and fructose. In the early stages of fruit development, these sugars synthesize starch, cellulose, hemicellulose, and various cellular components, or provide energy for cell division and rapid growth [39,40].

The sugar accumulation level in fruit is influenced by a variety of factors, such as the production, transportation, distribution, and metabolism of photosynthetic products in fruit, and the sugar metabolism process in fruit is extremely complex, and regulated by numerous enzymes [41,42,43,44]. Currently, the genes involved in sugar accumulation have been reported in many plants. The presence of *ClNAC68* in watermelon inhibits the expression of *ClINV* and the activity of transductase, thus preventing the breakdown of sucrose [45]. The overexpression of MdSWEET12a, a sucrose transporter protein, in apples and tomatoes caused an alteration in the expression of genes associated with sucrose metabolism and transport, leading to a rise in sucrose, glucose, and fructose contents [21]. Glucose is the main sugar in mature pitaya fruit, which is mainly regulated by vacuolar acid invertase and sucrose synthase [46]. We used whole-transcriptome sequencing techniques to discern discrepancies in transcriptomic dynamics among ‘Shushanggan’, ‘Sungold’, and ‘F43’ throughout three distinct phases. In this study, the sweetness of ‘Shushanggan’ fruits was determined using a combination of sucrose, fructose, glucose, and sorbitol contents, with genes like *SPP* (MTCONS_00064889), *E2.4.1.14* (PaF106G0100001858.01), *ENPP1_3* (MTCONS_00060314), *glgA* (MTCONS_00008488), *WAXY* (MTCONS_00047878), and *GBE1* (PaF106G0700027757.01) having the most significant impact, whereas previous research has indicated that the *ParSuSy5*, *ParSuSy6*, and *ParSuSy7* genes, as well as *ParFK1* fructokinase enzymes, may be essential for the accumulation of sugar in apricot fruit [13]. The content of organic acids in fruit ultimately depends on the equilibrium of acid production, decomposition, utilization, and distribution [34]. The metabolism of fruit acids is greatly affected by PEPC (phosphoenolpyruvate carboxylase), MDH (malate dehydrogenase), ME (malate enzyme), CS (citrate synthase), and ACO (aconitase) [35,36,47]. We found that MDH1s were differentially expressed at different developmental stages in the three apricot cultivars. Furthermore, the composition of the sugars and acids in fruits is influenced by a range of environmental elements (e.g., temperature, humidity, effective photosynthetic radiation, wind speed, rainfall, CO_2_ concentration, and duration of sunlight). We examined the data collected from sampling time points of three distinct apricot cultivars (Appendix A), and discovered that the levels of rainfall and CO_2_ remained consistent throughout these time intervals. Nevertheless, the accumulation of sugars and acids in apricot fruits may be influenced by factors such as air temperature, humidity, effective photosynthesis radiation, duration of sunlight, and wind speed.

WGCNA is an analytical method for analyzing gene expression patterns in multiple samples; it facilitates the grouping of genes with similar expression patterns, and the examination of the relationship between modules and phenotypes, and has been widely utilized in the investigation of phenotypic traits, and gene association analyses. An illustration of this could be that six hub genes, including *bHLH*, *MADS-box*, *C3H*, *GPA1*, *AHP*, and *EMB*, may have a strong correlation with kernel size when it comes to kernel consumption in apricot [48]. By building a network of co-expressed genes, Rohini Garg et al. have ascertained that the substantial size and weight of chickpea seeds were determined to occur via an extended duration of cell division during embryogenesis, heightened endoreduplication, and a greater accumulation of storage compounds during maturation [49]. We used WGCNA to explore a variety of structural genes, transcription factors, sugar transporter proteins, and other potential regulators linked to sugar and acid metabolism. Our findings revealed *VIP1* as a hub gene, along with four sugar transporter proteins (*SLC35B3*, *SLC32A*, *SLC2A8*, and *SLC2A13*), and three sugar metabolism genes (*MUR3*, *E3.2.1.67*, and *CSLD*) that interact with it. These eight genes exhibited significant expression at SS3 in ‘Shushanggan’ in contrast to ‘Sungold’ and ‘F43’, which were identified as crucial candidate genes for the positive regulation of sugar and acid metabolism in apricot. However, the mechanism by which *VIP1* interacts with sugar transporter and the sugar and acid metabolism genes remains to be further investigated.

Gene expression is regulated at multiple levels not only by transcription factors, histone modifications, and DNA methylation, but also by non-coding RNAs (ncRNAs). ncRNAs have been linked to a variety of physiological processes, with miRNA and lncRNA being especially influential in regulating transcription at different levels [50,51]. The development of high-throughput sequencing technology has led to the identification of ncRNAs from different species, and more and more studies have also shown that ncRNAs play an important role in plant growth, reproductive development, and fruit regulation [52,53,54,55]. To date, the functions of lncRNAs and miRNAs in the metabolism of sugar and acid in apricot fruit remain unidentified, and unreported. Our study is pioneering in exploring the impact of lncRNAs and miRNAs on the buildup of sugar and acid in apricot fruits, and it also strives to comprehend the regulatory mechanisms governing lncRNAs and miRNAs in sugar and acid metabolism. However, in the last few years, a great deal of research has been conducted to examine the influence of ncRNAs on fruit color and ripening. For example, Ran et al. discovered that *lncRNA1459* was associated with tomato fruit ripening, which had an effect on the production of ethylene in tomato fruits, and knocking out *lncRNA1459* using a CRISPR/Cas9 system can significantly slow down the ripening process of tomato [56]. The *MdBBX22-miR858-MdMYB9/11/12* module in apple regulates the production of proanthocyanidins [57]. Ma et al. found that when exposed to light in apple, *MdWRKY1* activates the expression of *MdLNC499* through sequence−specific interactions with its W−box element, and *MdLNC499* induces *MdERF109* to enhance anthocyanin accumulation by directly binding *MdERF109* to the *MdCHS*, *MdUFGT*, and *MdbHLH3* promoters [58]. During our investigation, we discovered 36 genes linked to sugar and acid metabolism that possess specific lncRNAs, while 34 genes exhibited specific miRNAs. In spite of the absence of comprehensive experimental validation, this study provides a basis for investigating the influence of lncRNAs and miRNAs on apricot sugar accumulation.

Taken together, we used apricots with varying levels of sugar (high, common, and low) to examine the alterations in sugar composition and content, as well as the expression of genes associated with sugar and acid metabolism during the growth of different apricot fruits, both in terms of physiology and molecules. The mechanism of sugar accumulation in apricot fruit was systematically analyzed, and the characteristics were pointed out during apricot fruit development. Regulating and reforming the sugar accumulation process offers a scientific foundation and technical methods to enhance the quality of apricot or cultivate new varieties.

## 4. Materials and Methods

### 4.1. Plant Material

Three apricot cultivars were used in this study, the high-sugar cultivar ‘Shushanggan‘, the common-sugar cultivar ‘Sungold’, and the low-sugar cultivar ‘F43′ (Figure 1A), and they were grown at the long-term experimental base of the Research Institute of Non-timber Forestry in Mengzhou City, Henan Province, China. Samples of apricot fruit were collected at three distinct stages of growth for the three cultivars, respectively: S1, green fruits with a hard kernel stage (SF1, 49 days after full bloom (DAFB); SJ1, 63 DAFB; SS1, 46 DAFB); S2, color-turning stage (SF2, 77 DAFB; SJ2, 74 DAFB; SS2, 67 DAFB); S3, commercial maturity stage (SF3, 98 DAFB; SJ3, 84 DAFB; SS3, 88 DAFB). The fruits of three cultivars were obtained at random from the three clones, with each replicate comprising ten fruits. After the apricot fruit was picked, the peel and pulp were separated with a scalpel, and the pulp was quickly put into a storage tube, frozen with liquid nitrogen, and stored at −80 °C. One part was used for the determination of sugar and acid content, and the other part was used for whole-transcriptome sequencing and qRT−PCR.

### 4.2. Sugar and Organic Acid Measurements

The soluble sugar and organic acid content in apricot fruit (fresh weight) were measured according to previous studies with some modifications [59]. We accurately weighed 2.00 g fruit in a 50 mL beaker, and added 25 mL 0.05% sulfuric acid solution to homogenize for 2 min. The homogenizer was cleaned with 10 mL 0.05% sulfuric acid solution, and we combined the cleaning solution. We transferred the extract to a 50 mL volume bottle, washed the inner wall of the beaker with 10 mL 0.05% sulfuric acid solution, combined the solution into a 50 mL volume bottle, and shook well with 0.05% sulfuric acid solution. After reaching 5 mL of constant volume, the sample was centrifuged (10,000 r/min, 4 °C) for 10 min, and the supernatant was taken through a 0.22 μm water needle filter to be measured. High-performance liquid chromatography (HPLC) via an Aminex HPX−87H column (7.8 mm × 300 mm) was used to determine the concentrations of sucrose, glucose, fructose, sorbitol, citric acid, and malate together with the following parameters: mobile phase = 0.1% sulfuric acid solution, column temperature = 35 °C, flow rate = 0.60 mL/min, injection volume = 10 μL, DAD detector wavelengths = 210 nm and 254 nm, RID detector temperature = 35 °C. Each experiment had three separate biological repetitions.

The content of starch was determined via an anthrone colorimetric method with reference to previous studies [60]. We weighed 2.0 g apricot pulp (fresh weight, duplicate/triplicate) in a 25 mL colorimetric tube, homogenized it with 80% ethanol, extracted sugar in a water bath at 80 °C, extracted 10 mL twice, removed the supernatant, and hydrolyzed the residue with 10 mL 3 mo/L hydrochloric acid. After the hydrolysis was completed, 10 mL 3 mol/L sodium hydroxide was added to neutralize it to a neutral constant volume using a scale. We divided 0.2 mL + 2.3 mL water into 10 mL colorimetric tubes, and added 6.5 mL anthrone solution at 620 nm for colorimetric comparison. Using glucose solution as the standard curve. We determined that titratable acid in apricot fruit with reference to GB12456−2021.

### 4.3. Total RNA Extraction, lncRNA and Small RNA Library Construction, and Sequencing

Total RNA was extracted using a FastPure^®^ Universal Plant Total RNA Isolation Kit (Vazyme Biotech Co., Ltd., Nanjing, China). Three biological replicates were analyzed for each RNA sample, except for SF2 and SS2, with two replicates (the quality of SF2_3 and SS2_1 RNA was too low to be analyzed). A ribosome-chain-specific model was used to construct a long non-coding RNA library, the PE100 mode was used for sequencing on a Zebra platform (BGI−Shenzhen, Shenzhen, China), and 10G clean data were obtained for each sample. The short reads comparison tool SOAP (version: v1.5.2) [61] was used to compare reads to the ribosome database for data filtering. Clean reads were aligned to the apricot reference genome (Genome Database for Rosaceae, tfGDR1049) using the alignment software HISAT (version: v2.0.4) [62], followed by alignment and assembly with the strain-specific patterns of HISAT [62] and StringTie (version: v1.0.4) [63]. Cuffcompare (version: v2.2.1) [64] was used to compare the assembled transcripts with known mRNAs and lncRNAs to obtain information about their mutual position relationships. Small RNA libraries were constructed by purifying total RNA into 18–30 nt RNA using PAGE electrophoresis gel, and qualified libraries were sequenced using the BGISIQ–500 platform (BGI-Shenzhen, Shenzhen, China). miRNA-based precursors were able to form hairpin secondary structures, and new miRNA predictions were made using miRanda (version: 3.3a) [65]. The RNA sequencing raw data were submitted to the Genome Sequence Archive (PRJNA1031355).

### 4.4. Differentially Expressed Genes (DEGs) and Enrichment Analysis

The differential analysis software DEGseq (1.56.1) [66] was used to compare the gene expression levels of the two groups of samples to identify the differentially expressed genes (DEGs). In this study, according to the filter conditions of DEGs (Log2 (fold-change) ≥ 2.00 and Adjusted *p*-value ≤ 0.001), transcripts including coding RNA and non-coding RNA were differentially expressed transcripts. To reduce transcription noise, we considered gene expressed if its average FPKM value was > 5 for further analysis. To quantitatively evaluate whether the candidate genes (differential mRNA and differential lncRNA target genes) of a certain functional module were enriched, we used a hypergeometric model to calculate the enrichment of candidate genes for each functional module. Each functional module corresponds to a *p*-value. The smaller the *p*-value, the more abundant the gene on this functional module. Then, we performed FDR correction on the *p*-value, and we set FDR ≤ 0.01 functions as indicating significant enrichment. The q-value (the q-value was corrected using the Bonferroni [67] method) ≤ 0.05 pathway was defined as a KEGG pathway with significant differences in miRNA target gene expression.

### 4.5. Target Gene Prediction of lncRNAs and miRNAs

The functions of lncRNAs are mainly achieved via cis or trans acting on the target genes. The prediction principle of cis acting on the target genes considers that the function of lncRNAs are related to the protein coding gene near its coordinates, so mRNAs (lncRNA 10 k upstream or 20 k downstream of mRNA) adjacent to lncRNAs are selected as target genes. As trans regulation is not dependent on the location relationship, we predict it by combining energy methods. RNAplex [68] was used to analyze the binding energy of lncRNAs and mRNAs. If the binding energy is < −30, it is determined as a trans relationship. The potential target genes of miRNAs were predicted by combining the free energy and miRNA score using psRobot (version: v1.2) [69], TAPIR (https://www.vandepeerlab.org/?q=tools/tapir (accessed date 31 October 2023)) [70] and TargetFinder [71] software (https://targetfinder.org/ (accessed date 31 October 2023)).

### 4.6. WGCNA and Gene Network Visualization

A total of 7993 significant DEGs from two cultivars were used to perform weighted gene co-expression network analysis (WGCNA) using the WGCNA in R [72]. The modules were obtained using the automatic network build function block with default settings. The soft power was 20, the min module size was 30, and the module cuttree height was 0.4. The correlation-based associations between phenotypic data (sucrose, fructose, glucose, sorbitol, malate, citric acid, and starch content) and gene modules were calculated using the default settings. To identify the relationship between the phytohormone-related structural genes and TF genes, differentially expressed genes with module membership (|MM| > 0.8) and gene significance (|GS| > 0.2) were defined as hub genes with potentially important functions. The major co-expression networks for each significant module based on the weight values (top 100) and degree of connectivity between the above-selected candidate genes were exported using Cytoscape v3.7.2.

### 4.7. Real-Time Quantitative PCR

Total RNA was extracted using a FastPure^®^ Universal Plant Total RNA Isolation Kit, and reverse transcription was performed using a RevertAidTM First-Strand cDNA Synthesis Kit with DNase I (Vazyme Biotech Co., Ltd., Nanjing, China). All responses were obtained by applying 2 × SYBR Green qPCR Premix (Universal) (Vazyme Biotech Co., Ltd., Nanjing, China) in a whole sample of 20 μL (2 × SYBR Green qPCR Premix: 10 μL; primers: 0.4 μL; cDNA: 1 μL; and ddH2O: 8.2 μL). The ubiquitin (UBQ) gene [73] was used as an internal reference gene to verify qRT−PCR, and the relative expression of the genes was calculated according to the 2^−ΔΔCT^ method [74]. The Pearson correlation coefficients and statistical significance between the fold changes among different samples from qRT−PCR and RNA−seq were calculated using SPSS 26.0. Primers were designed using Primer3 (https://primer3.ut.ee/ (accessed on 1 August 2012)), and specific information is listed in Appendix A.

### 4.8. Statistical Analysis and Plotting

In analyzing the differences in sugar content among the three apricot varieties, a line chart of the significance of sugar and organic acid, determined via a Student’s *t*-test, was plotted using the GraphPad Prism8 v8.0.2 software. Venn maps of DEGs in the three growth and development stages of the three apricot cultivars and a heat map of DEGs in sugar metabolic pathways were drawn usinh TBtools v2.0 software. The Pearson correlation coefficients (r) and statistical significance (p) between RNA−seq and qRT−PCR were evaluated based on their respective mean values at each stage using SPSS v26.0.1.0 software. Unless otherwise noted, figures used to display the statistics were plotted using GraphPad Prism8 v8.0.2 software.

## 5. Conclusions

In conclusion, comprehensive metabolomic and transcriptomic analyses were conducted during fruit growth and development in three apricot cultivars with contrasting fruit sugar content. We found high sucrose, fructose, glucose, and sorbitol content in high-sugar apricots, while low-sugar apricots accumulated more titratable acid, citric acid, and starch. We comprehensively analyzed the details of expression differences in DEGs among the three apricot cultivars that may affect sugar and acid metabolism. Our results suggest that *VIP* transcription factors, sugar transporters, starch, and sucrose metabolic pathway genes, and their lncRNAs and miRNAs, play a crucial role in the transcriptional regulation of sugar accumulation, which is why ‘Shushanggan’ apricots are relatively sweet. These results provide a clue for exploring the functional genes related to sugar metabolism and lay a foundation for the selection and cultivation of almond high-sugar apricot fruits. In addition, the results of this study also provide a basis and inspiration for high-sugar apricot variety breeding, which must still be considered from the perspective of sucrose, fructose, glucose, sorbitol, and other substances combined.

## Figures and Tables

**Figure 1 ijms-24-16992-f001:**
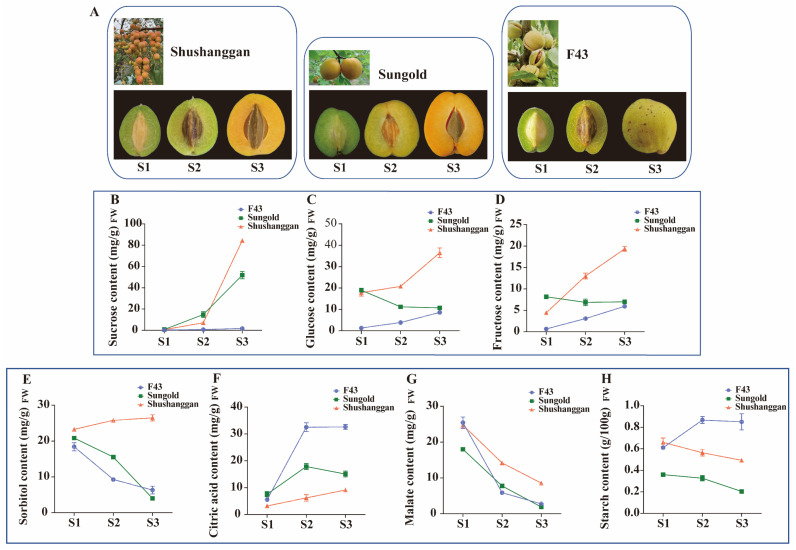
Changes in soluble sugar and organic acid contents at different developmental stages of apricot. (**A**) Phenotype of three apricots. (**B**–**H**) The contents of sucrose (**B**), glucose (**C**), fructose (**D**), sorbitol (**E**), citric acid (**F**), malate (**G**), and starch (**H**).

**Figure 2 ijms-24-16992-f002:**
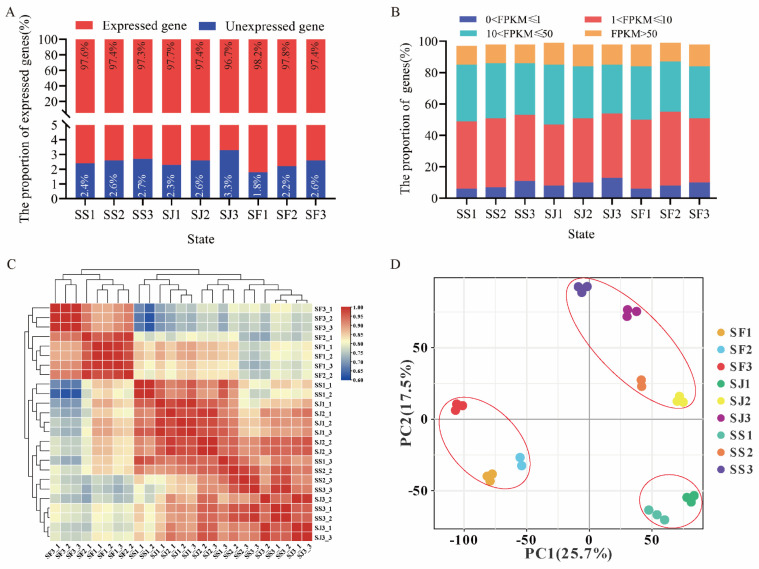
Gene expression and correlation between the transcriptomes of three representative stages each of three apricot cultivars. (**A**) The proportion of expressed genes in three apricot cultivars of three typical stages. (**B**) The proportion of expressed genes at four different expression levels in ‘Shushanggan’ (SS1−3), ‘Sungold’ (SJ1−3), and ‘F43’ (SF1−3). (**C**) Pearson correlation coefficient (PCC) of expressed genes from three representative stages in ‘Shushanggan’, ‘Sungold’, and ‘F43’. (**D**) Principal component analysis (PCA) plot showing clustering of transcriptomes of three representative stages in ‘Shushanggan’, ‘Sungold’, and ‘F43’.

**Figure 3 ijms-24-16992-f003:**
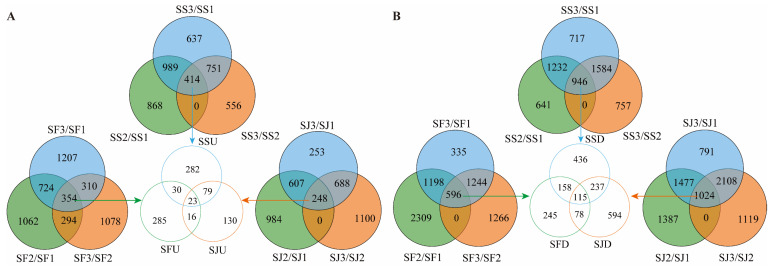
Identification of candidate genes correlated with sugar and acid metabolism in ‘Shushanggan’, ‘Sungold’, and ‘F43’ fruits. (**A**) SSU, SJU, SFU: genes upregulated in ‘Shushanggan’, ‘Sungold’, and ‘F43’, respectively. (**B**) SSD, SJD, SFD: genes downregulated in ‘Shushanggan’, ‘Sungold’, and ‘F43’, respectively.

**Figure 4 ijms-24-16992-f004:**
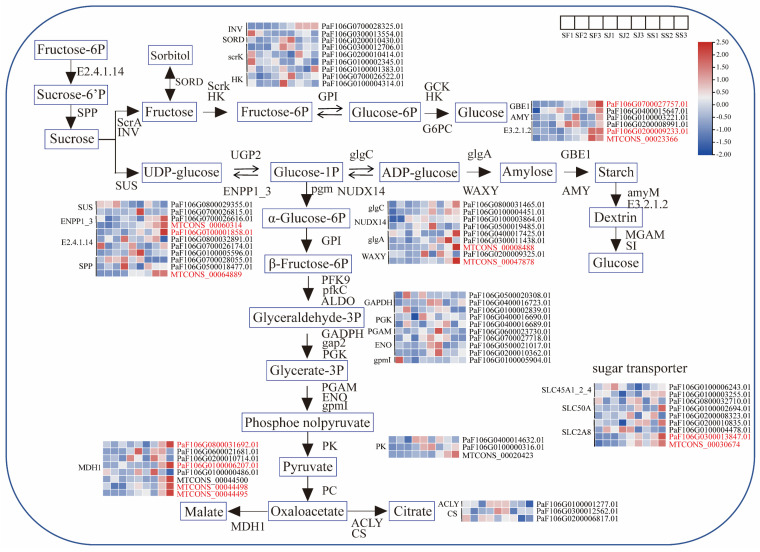
Expression patterns of DEGs involved in sugar and acid metabolic pathway. E2.4.1.14, sucrose-phosphate synthase; SPP, sucrose-6-phosphatase; ScrA, sucrose PTS system EIIBCA or EIIBC component; INV, beta-fructofuranosidase; SORD, L-iditol 2-dehydrogenase; ScrK, fructokinase; HK, hexokinase; GPI, glucose-6-phosphate isomerase; GCK, glucokinase; G6PC, glucose-6-phosphatase; SUS, sucrose synthase; UGP2, UTP-glucose-1-phosphate uridylyltransferase; ENPP1_3, ectonucleotide pyrophosphatase family member 1/3; glgC, glucose-1-phosphate adenylyltransferase; NUDX14, ADP-sugar diphosphatase; glgA, starch synthase; WAXY, granule-bound starch synthase; GBE1, 1,4-alpha-glucan branching enzyme; AMY, alpha-amylase; amyM, maltogenic alpha-amylase; MGAM, maltase−glucoamylase; SI, sucrase-isomaltase; E3.2.1.2, beta-amylase; pgm, phosphoglucomutase; PFK9, 6-phosphofructokinase; pfkC, ADP-dependent phosphofructokinase/glucokinase; ALDO, fructose-bisphosphate aldolase, class I; GADPH, glyceraldehyde 3-phosphate dehydrogenase; gap2, glyceraldehyde-3-phosphate dehydrogenase (NAD(P)+) (phosphorylating); PGK, phosphoglycerate kinase; PGAM, 2,3-bisphosphoglycerate-dependent phosphoglycerate mutase; ENO, enolase; gpmI, 2,3-bisphosphoglycerate-independent phosphoglycerate mutase; PK, pyruvate kinase; PC, pyruvate carboxylase; ACLY, ATP citrate (pro−S)-lyase; CS, citrate synthase; MDH1, malate dehydrogenase.

**Figure 5 ijms-24-16992-f005:**
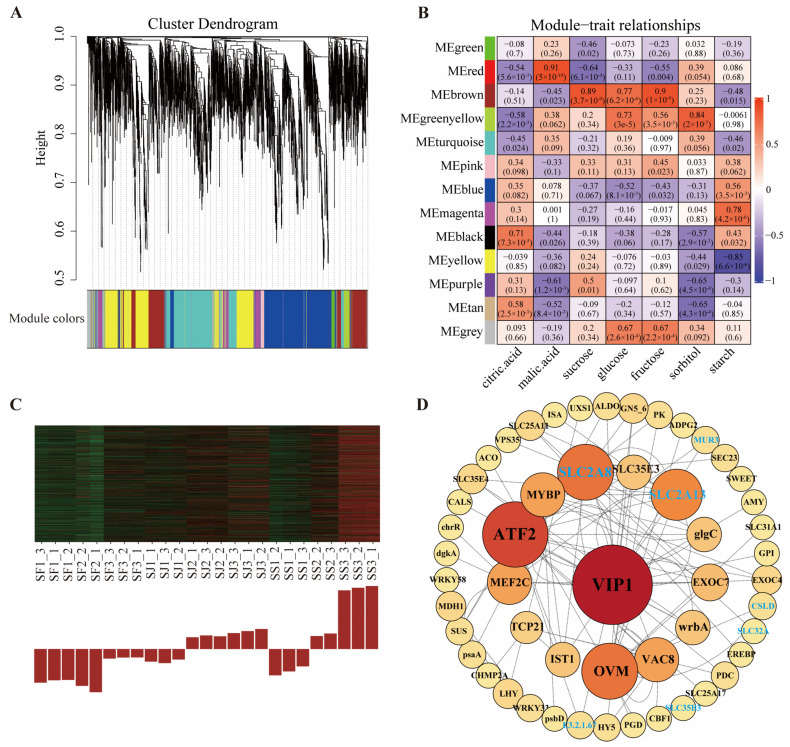
Identification of hub genes in co-expression network. (**A**) Hierarchical clustering tree (cluster dendrogram) illustrating 13 modules of co-expressed genes pursuant to WGCNA. The same color indicates that the corresponding gene on the clustering tree belongs to the same module. (**B**) heatmap of module−sugar and acid relationships. Each row represents a module indicated by different colors. The color key from blue to red represents correlation values from −1 to 1. (**C**) MEbrown module gene expression patterns. (**D**) The co-expression network contained 52 co-expression genes for the MEbrown module.

**Figure 6 ijms-24-16992-f006:**
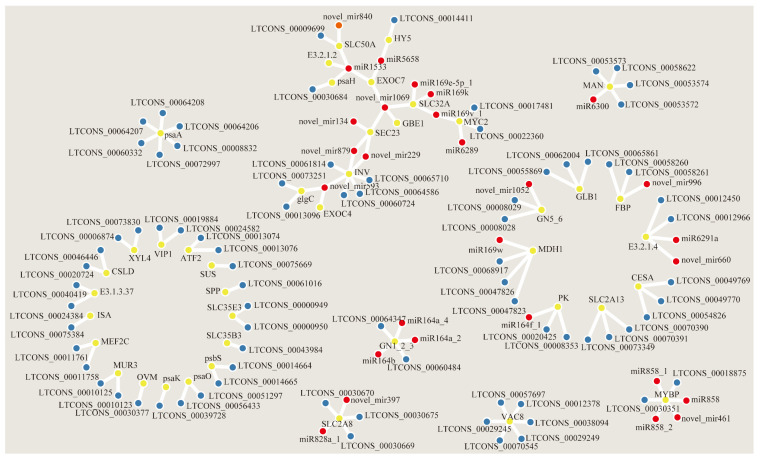
Interaction networks of differential expression of sugar accumulation genes with lncRNAs and miRNAs. Yellow, blue, and red circles indicate mRNA, lncRNA, and miRNA, respectively.

**Figure 7 ijms-24-16992-f007:**
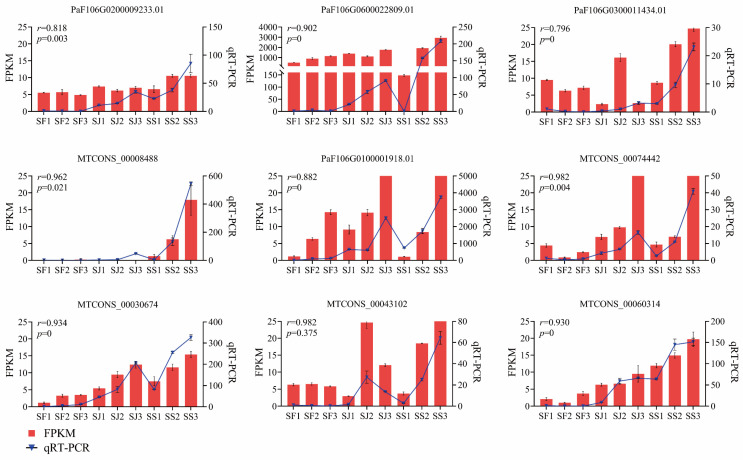
Expression levels of 9 candidate DEGs validated via qRT−PCR. Columns and lines indicate RNA−seq and qRT−PCR of the candidate DEGs, respectively. Pearson correlation coefficients were calculated between qRT−PCR and RNA−Seq data of candidate DEGs. UBQ was used as the internal control. Error bars indicate SD.

## Data Availability

The raw transcriptome sequencing reads in this research were uploaded to the NCBI SRA database (PRJNA1031355).

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
