# Peer review of "Transcriptome and Metabolome Analyses Reveal Sugar and Acid Accumulation during Apricot Fruit Development"

_ijms, 2023, doi:10.3390/ijms242316992_

Round 1
Reviewer 1 Report
Comments and Suggestions for Authors
This manuscript by Ningning Gou and colleagues covers an interesting topic related with sugar and acid accumulation during apricot maturation and their relationship with metabolic pathway.
In general, the research work is correct and understanding to the readers. Authors show a great number of valuable results. However, in my opinion, the work could be improve introducing a few changes in several points before a possible publication.
1 – Line 92: It is not clear what is meant by "high-sugar cultivars", "common sugar cultivars" and "low-sugar cultivars". The authors must present an objective classification, for example with a range of values and possibly references that support this classification.
2 – Line 95: “A significant difference”. However, authors didn’t show in figure 1 any statistical analysis to prove if the differences are significant different or not.
3 – Line 99 – 100 and during all work. It is not clear if the results are show based on fresh or dry weight. I suggest clarifying.
4 - To also support the dynamics of accumulation of sugars and acids, it would be interesting to have climate data (temperature and rainfall). The evolution of these compounds is also affected by environmental factors.
5 – pH and total acidity results, could be useful to add.
6 – Line 372-379: These sentences could be introduced in conclusions.
7 – Line 396-409: How were the different sugars identified by HPLC? Were the determinations carried out in duplicate/triplicate ?
8 – I suggest rewriting the conclusions. Authors are repeating the results.
Author Response
November 23, 2023
Editorial Board
International Journal of Molecular Sciences
Dear editors and reviewers,
We sincerely appreciate your careful editing and comments concerning our manuscript, entitled “Transcriptome and Metabolome Analysis Reveal Sugar and Acid Accumulation during Apricot Fruit Development (ijms-2722373)”. Here, the revised version of the manuscript is submitted. We have carefully addressed all reviewer comments in the current revised version of the manuscript. We list the comments and our responses (in blue) below. We wish to thank the reviewers for their careful reading of the manuscript and their many useful suggestions.
We hope the revised manuscript is now acceptable for publication in your prestigious journal, and hope to hear from you soon.
Sincerely,
Ningning Gou
Response to Reviewer 1 Comments
Point 1: Line 92: It is not clear what is meant by "high-sugar cultivars", "common sugar cultivars" and "low-sugar cultivars". The authors must present an objective classification, for example with a range of values and possibly references that support this classification.
Response 1: Thank you for this important comments. In the revised manuscript, we have added total soluble sugar content of apricot fruits from different cultivars to support this classification for ‘high-sugar cultivars’, ‘cultivars sugar cultivars’, and ‘low-sugar cultivars’ in line 90-92 of result 2.1: ”We determined the content of total soluble sugar (TSS) in different apricot fruits, defining TSS > 11% as high-sugar cultivars, 8%< TSS < 10% as common sugar cultivars , and TSS < 3% as low-sugar cultivars (Supplementary FigureS1, Table S1)”.
Point 2: Line 95: ”A significant difference”. However, authors didn’t show in figure 1 any statistical analysis to prove if the differences are significant different or not.
Response 2: Thank you for this suggestion. In the revised manuscript, we provided statistic analysis data in Supplementary Table S2 for sucrose, glucose, fructose, sorbitol, citric acid, malate, starch, titratable acid and pH at S1, S2 and S3 periods among ‘shushanggan’, ‘sungold’ and ‘F43’.
Point 3: Line 99-100 and during all work. It is not clear if the results are show based on fresh or dry weight. I suggest clarifying.
Response 3: Thank you for this suggestion. Our results are based on the fresh weight of apricot fruits. In the revised manuscript, we have revised the description in line 423 of materials and methods section and signed FW in Figure 1B, C, D, E, F, G, H which could be more rigorous.
Point 4: To also support the dynamics of accumulation of sugars and acids, it would be interesting to have climate data (temperature and rainfall). The evolution of these compounds is also affected by environmental factors.
Response 4: Thanks for this important suggestion. We agree with this comment that the accumulation of sugars and acids is influenced by a combination of environmental factors, and in the revised manuscript we have added descriptions of climate data in line 342-349: “Furthermore, the composition of sugars and acids in fruits is influenced by a range of environmental elements (e.g. temperature, humidity, effective photosynthetic radiation, wind speed, rainfall, CO2 concentration, and duration of sunlight). We examined the data collected from sampling time points of three distinct apricot varieties (Table S5) and discovered that the levels of rainfall and CO2 remained consistent throughout these time intervals. Nevertheless, the accumulation of sugars and acids in apricot fruits may be influenced by factors such as air temperature, humidity, effective photosynthesis radiation, duration of sunlight, and wind speed”.
Point 5: pH and total acidity results, could be useful to add.
Response 5: Thank you for this important suggestion. We have supplemented pH and titratable acid data to line 115-118 of result 2.1: “In addition, we measured titratable acid and pH and found that titratable acid was lower in ‘shushanggan’ than ‘sungold’ was lower than ‘F43’ and had a pH greater than ‘F43’ greater than ‘sungold’ (Supplementary FigureS2)”.
Point 6: Line 372-379: These sentences could be introduced in conclusions.
Response 6: Thank you for this suggestion. In the revised manuscript, we have added this to the conclusion.
Point 7: Line 396-409: How were the different sugars identified by HPLC? Were the determinations carried out in duplicate/triplicate ?
Response 7: Thanks for your questions and comments. All samples were tested in duplicate/triplicate. In the revised manuscript, using HPLC to identify the different sugars and acids we have described in detail in line 423-437 of materials and methods.
Point 8: I suggest rewriting the conclusions. Authors are repeating the results.
Response 8: Thanks for this comment. In the revised manuscript, we have rewritten the conclusion.

Reviewer 2 Report
Comments and Suggestions for Authors
When the language and format is fixed I think this is a nice article for the journal. It is parts with not understandable language. It is typing errors. The objective is not typically written, it would be better to have some of that in the abstract.
Author Response
November 23, 2023
Editorial Board
International Journal of Molecular Sciences
Dear editors and reviewers,
We sincerely appreciate your careful editing and comments concerning our manuscript, entitled “Transcriptome and Metabolome Analysis Reveal Sugar and Acid Accumulation during Apricot Fruit Development (ijms-2722373)”. Here, the revised version of the manuscript is submitted. We have carefully addressed all reviewer comments in the current revised version of the manuscript. We list the comments and our responses (in blue) below. We wish to thank the reviewers for their careful reading of the manuscript and their many useful suggestions.
We hope the revised manuscript is now acceptable for publication in your prestigious journal, and hope to hear from you soon.
Sincerely,
Ningning Gou
Response to Reviewer 2 Comments
Point 1: When the language and format is fixed I think this is a nice article for the journal. It is parts with not understandable language. It is typing errors. The objective is not typically written, it would be better to have some of that in the abstract.
Response 1: Thank you for this important suggestion. We are very sorry for our unintelligible language writing. we have carefully checked and improved the English writing in the revised manuscript.

Reviewer 3 Report
Comments and Suggestions for Authors
Positive notes to the authors:
1. The topic of the manuscript is relevant as it is related to Transcriptome and Metabolome Analysis Reveal Sugar and Acid Accumulation during Apricot Fruit Development;
2. The authors have explored the level of biochemical plant genetics up to the time of conducting their studies and have upgraded the science in this field;
3. The manuscript mainly focuses on the metabolism of fruit sugar and organic acids, as well as management of this metabolism depending on the development of the apricot fruit;
4. The results are relatively well illustrated with the help of color photos. The figures, however, are indistinct and should be enlarged by 15-20% for better visualization;
5. The discussion is relatively complete and concerns the biochemical genetics of the apricot and the regulation of carbohydrate and acid metabolism. The results of other scientists in the relevant field are also considered;
6. In the Material and methods section, the methods and the processing and research of the biological material by the authors of the manuscript are described in detail.;
7. Conclusions are drawn as a result of the study regarding carbohydrate and acid metabolism, and future research with new varieties of apricots is foreseen.
Negative notes and recommendations to the authors:
1. I recommend zooming in figures 1, 3, 4, 5, 6 and 7 by 10-15% for better visualization;
2. The Discussion section shows the erudition of the authors of each scientific article. In this aspect, I recommend that the authors dig deeper into the bowels of biochemical genetics to reveal the intimate mechanisms of carbohydrate and acid metabolism;
3. In the Conclusions section, there are no recommendations to the fruit growing branch regarding improving the breeding of sweet apricot varieties by the agronomists.
Comments on the Quality of English LanguageNotes on English quality:
The manuscript is written in relatively good English without linguistic and grammatical errors. Still, I recommend a final polish by an English-speaking editor.
Author Response
November 23, 2023
Editorial Board
International Journal of Molecular Sciences
Dear editors and reviewers,
We sincerely appreciate your careful editing and comments concerning our manuscript, entitled “Transcriptome and Metabolome Analysis Reveal Sugar and Acid Accumulation during Apricot Fruit Development (ijms-2722373)”. Here, the revised version of the manuscript is submitted. We have carefully addressed all reviewer comments in the current revised version of the manuscript. We list the comments and our responses (in blue) below. We wish to thank the reviewers for their careful reading of the manuscript and their many useful suggestions.
We hope the revised manuscript is now acceptable for publication in your prestigious journal, and hope to hear from you soon.
Sincerely,
Ningning Gou
Response to Reviewer 3 Comments
Point 1: I recommend zooming in figures 1, 3, 4, 5, 6 and 7 by 10-15% for better visualization;
Response 1: Thank you for your important suggestion. In the revised manuscript, we have zoomed all the figures in the text for optimal visualization.
Point 2: The discussion section shows the erudition of the authors of each scientific article. In this aspect, I recommend that the authors dig deeper into the bowels of biochemical genetics to reveal the intimate mechanisms of carbohydrate and acid metabolism;
Response 2: Thank you for this suggestion. We agree with this and further discuss the intrinsic mechanisms of acid metabolism and the expression of acid metabolism genes in our results in the revised manuscript in line 336-342: “The amount of organic acids present in fruit is ultimately determined by the equilibrium of acid production, breakdown, utilization, and distribution[47]. The metabolism of fruit acids is significantly impacted by PEPC (phosphenolpyruvate carboxylase), MDH (malate dehydrogenase), ME (malate enzyme), CS (citrate synthase) and ACO (aconitase) [36, 48, 49]. We found that MDH1 was differentially expressed at different developmental stages in the three apricot cultivars”.
Point 3: In the Conclusions section, there are no recommendations to the fruit growing branch regarding improving the breeding of sweet apricot varieties by the agronomists.
Response 3: Thank you for your important suggestion. In the revised manuscript, we have included this recommendation in line 542-544 of the concluding section. “In addition, the results of this study also provide the basis and inspiration, high sugar apricot variety breeding must still be considered from the perspective of sucrose, fructose, glucose, sorbitol and other substances combined contributions”.
